# ED2RM: Equivariant Denoising Diffusion Models based on Riemannian Morphological PDEs

## Abstract

Diffusion models have recently emerged and demonstrated remarkable capabilities in high-quality image synthesis and data generation. This work addresses two key issues in recent Denoising Diffusion Probabilistic Models (DDPMs), inspired by nonequilibrium thermodynamics: geometric feature extraction and equivariance. To tackle these challenges, we introduce a geometric approach to the prediction network of DDPMs by designing equivariant morphological partial differential equations (PDEs) for group convolutional neural networks (G-CNNs), referred to as PDE-G-CNNs. These PDEs are formulated on Riemannian manifolds to better capture nonlinearities, represent thin geometrical structures, and incorporate symmetries into the learning process. Our method achieves this by considering a system of two PDEs: a convection term and a first-order Hamilton–Jacobi-type PDE that acts as morphological multiscale dilations and erosions. Preliminary experiments on MNIST and RotoMNIST indicate significant performance gains compared to baseline DDPMs.

## 1 Introduction

In recent years, deep generative models have experienced rapid growth, with applications ranging from realistic image generation Goodfellow et al. (2014); Kingma & Welling (2013); Kingma et al. (2014); Dhariwal & Nichol (2021); Ho et al. (2020) to audio synthesis Chen et al. (2020); Popov et al. (2021), and even molecular modeling Simonovsky & Komodakis (2018); Gebauer et al. (2019); Simm et al. (2020); Hoogeboom et al. (2022). Among these approaches, diffusion probabilistic models (DPM) Sohl-Dickstein et al. (2015); Song & Ermon (2019); Ho et al. (2020); Song et al. (2021); Croitoru et al. (2023) have emerged as particularly influential with impressive generative capabilities. DPM can be grouped into three broad categories: denoising DPM (DDPM) Sohl-Dickstein et al. (2015); Ho et al. (2020) inspired by the theory of nonequilibrium thermodynamics, noise conditioned score networks (NCSNs) Song & Ermon (2019) based on generative models through a multi-scale denoising score matching objective, and stochastic differential equations (SDEs) Song et al. (2021); Huang et al. (2021). In particular, in the field of image generation, DDPM Ho et al. (2020) have demonstrated a remarkable ability to produce high-quality samples. Their principle relies on two complementary steps. The first, known as the forward diffusion process, consists in progressively adding Gaussian noise to the data until their distribution approaches an isotropic normal law. The second, the reverse or denoising process, aims to invert this procedure by learning, through a deep neural network, the noise that must be removed in order to reconstruct the original data. Training is performed within a probabilistic framework, by optimizing a variational lower bound (ELBO) of the likelihood, which ensures the theoretical soundness of the model. Compared to other families of generative models, such as variational autoencoders Kingma & Welling (2013); Rezende et al. (2014) or generative adversarial networks (GANs) Goodfellow et al. (2014); Goodfellow (2017), DPMs stand out for the stability of their training and the diversity of the generated samples. DPM had been extended in Riemannian manifolds by deriving a Riemannian continuous-time ELBO Huang et al. (2022). A Riemannian extension of DDPM have been recently proposed for learning distributions on submanifolds of $\mathbb{R}^n$ Liu et al. (2025). Score-based matching models have also been extended to Riemannian manifolds Bortoli et al. (2022). A generalized strategy for

numerically computing the heat kernel on Riemannian symmetric spaces in the context of denoising score matching was also proposed Lou et al. (2023).

Equivariance plays an important role in most neural networks architectures. Equivariance means that the operation of performing a transformation of the input data then passing them through the network is the same as passing the input data through the network and then performing a transformation of the output. It can be used to learn the symmetries in data. Such a principle has recently been used for molecule generation by combining $E(n)$ equivariant graph neural networks (EGNNs) Satorras et al. (2021) and the equivariance in $E(3)$ of the denoising distribution in the diffusion process of DDPM Hoogeboom et al. (2022). A similar equivariant approach was proposed for 3D molecule generation Cornet et al. (2024) with a learnable forward process. An equivariant diffusion model was also proposed Brehmer et al. (2023) with a $SE(3) \times \mathbb{Z} \times S_n$ invariance of distribution over trajectories. An $E(3)$ equivariant model was proposed Igashov et al. (2024) with an $O(3)$ invariance in the conditional diffusion model, which was used for molecular linker design. Deep neural networks are inherently invariant under translation transformations. To extend this invariance to other types of transformations, group-convolution (G-CNN) were introduced Cohen & Welling (2016); Bekkers et al. (2018); Cohen et al. (2019) and generalize CNNs so that symmetries are integrated during the learning process. G-CNN were shown to noticeably improved traditional CNN Winkels & Cohen (2018); Cohen et al. (2018); Bekkers (2019). A PDE framework, termed PDE-G-CNN, was recently introduced Smets et al. (2022); Bellaard et al. (2023) as a generalization of G-CNN. In Diop et al. (2024), equivariant PDE-G-CNN were integrated into GAN models and had shown significant gains in the quality of sample generation, as well as an increase of the robustness to data under geometric transformations.

**Contributions** Contrary to existing Riemannian extensions of DPMs, we propose here a proper Riemannian extension of DDPM by considering a Riemannian manifold endowed with a general Riemannian metric. Also, the equivariance property is introduced differently. We consider here a Lie group as the group of symmetries in order to take advantage of the group structure, on the one hand, and to benefit from the Riemannian metric, on the other hand. We summarize our main contributions as follows:

- Introduction of an equivariant denoising diffusion model (ED2RM) that integrates morphological PDEs into the prediction network. (See Fig. 1)

- Design of a diffusion model that is equivariant to translations, rotations, reflections, and permutations, ensuring that data symmetries are preserved throughout the learning process.

- Construction of a prediction network based on PDE-G-CNNs defined on Riemannian manifolds, which allows the model to better capture nonlinearities and fine geometric structures.

- Geometric interpretability of the proposed ED2RM framework, as its operations correspond to well-understood PDEs such as convection and Hamilton–Jacobi equations for multiscale dilations and erosions.

- Improvement of key geometric feature extraction within the denoising step, enhancing the overall robustness of the model.

- Improvement in the quality of image generation, demonstrated through experiments on MNIST and RotoMNIST, where ED2RM shows faster convergence and better FID scores compared to baseline DDPMs.

## 2 PRELIMINARIES

### 2.1 PROBABILISTIC DIFFUSION MODELS

Diffusion models are generative models based on the progressive addition of random noise to real data, followed by learning the inverse process that removes this noise (denoising) using a neural network. Additional details on this section are provided in Appendix A.

**Forward Process.** The forward process gradually adds noise to the variables $n_t$, for $t \in \{0, \ldots, T\}$, i.e., $n_{t+1} = n_t + \text{noise}$, with random noise. A conditional distribution then models

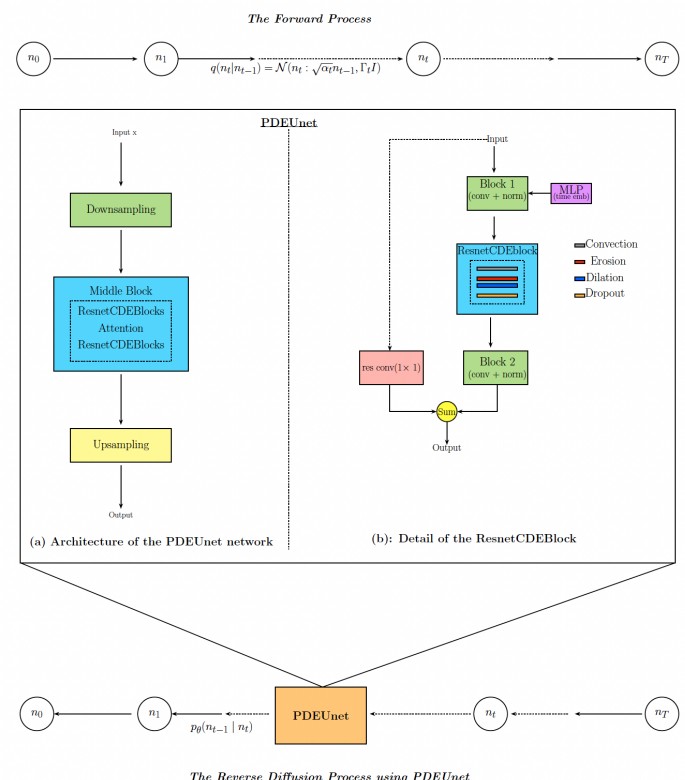

Figure 1: Our ED2RM approach uses PDEs-based equivariant layers in the denoising network. These layers enforce symmetry with respect to translations, rotations, reflections, and permutations, while also improving the extraction of fine geometric structures.

the probability of obtaining $n_{t+1}$ given $n_t$, denoted $q(n_{t+1} \mid n_t)$, which follows the Gaussian distribution:

$$q(n_t \mid n_{t-1}) = \mathcal{N}(n_t : \sqrt{\alpha_t}n_{t-1}, \Gamma_t I) \tag{1}$$

where $\alpha_t$ is defined according to the variance-preserving process proposed by Ho et al. (2020), i.e., $\alpha_t = 1 - \Gamma_t$. This parameter controls the amount of signal retained, and $\Gamma_t \in (0, 1)$ represents the noise level added at each step $t$. It acts progressively, so that the mean $\alpha_t n_{t-1}$ increasingly deviates from the already noised data $n_{t-1}$.

**Inverse Generative Process.** The diffusion process, or inverse generative process, generates data progressively from noise by following the true denoising process, denoted $P(n_{t-1} \mid n_t)$, which defines the probability of obtaining $n_{t-1}$ from $n_t$. This distribution is Gaussian, similarly to the forward process.

Since $x_0$ is unknown during denoising, a neural network $\phi$ parameterized by $\theta$ is used to approximate the inverse Gaussian conditional distribution, denoted $p_\theta(n_{t-1} \mid n_t)$, defined as:

$$p_\theta(n_{t-1} \mid n_t) = \mathcal{N}(n_{t-1} : \mu_\theta(n_t, t), \Sigma_\theta(n_t, t)) \tag{2}$$

where $\mu_\theta \in \mathbb{R}^d$ and $\Sigma_\theta \in \mathbb{R}^{d \times d}$ represent the mean and covariance matrix of the distribution at iteration $t$, respectively. For simplicity, as proposed in Ho et al. (2020), we set $\Sigma_\theta(n_t, t) = \sigma_t^2 I$, with constants $\sigma_t^2$ that depend on time but are not learned.

**Variational Lower Bound of the Likelihood.** The optimization problem for the Evidence Lower Bound (ELBO) with respect to $\theta$ is given by:

$$\underset{\theta}{\text{minimize}} \quad \sum_{t=2}^{T} \text{KL}(q(n_{t-1} \mid n_t, n_0) \parallel p_\theta(n_{t-1} \mid n_t)) - \mathbb{E}_{q(n_{1:T}|n_0)}[\log p_\theta(n_0 \mid n_1)]. \tag{3}$$

This equation trains the inverse distribution $p_\theta(n_{t-1} \mid n_t)$ to match the true denoising distribution $q(n_{t-1} \mid n_t, n_0)$ by minimizing their KL divergence. It can therefore be used as a loss function for a neural network parameterized by $\theta$, emphasizing the alignment between these two distributions. Details leading to Equation 3 are provided in Appendix A.

## 2.2 Equivariance

**Definition 2.1** *Let $G$ be a connected Lie group with identity element $e$ and $(\mathcal{M}, \mathbf{g})$ a connected Riemannian manifold $\mathcal{M}$ with metric $g$. A left action of $G$ on $(\mathcal{M}, \mathbf{g})$ is a map $\varphi : G \times (\mathcal{M}, \mathbf{g}) \to (\mathcal{M}, \mathbf{g})$ satisfying:*

*1. $\varphi(e, x) = x$, $\forall\, x \in (\mathcal{M}, \mathbf{g})$.*

*2. $\varphi(g, \varphi(h, x)) = \varphi(gh, x)$, $\forall\, g, h \in G$ and $\forall\, x \in (\mathcal{M}, \mathbf{g})$.*

Let $\varphi : G \times (\mathcal{M}, \mathbf{g}) \to (\mathcal{M}, \mathbf{g})$ be a left action of $G$ on $(\mathcal{M}, \mathbf{g})$. For a fixed $g \in G$, we define $\varphi_g : (\mathcal{M}, \mathbf{g}) \to (\mathcal{M}, \mathbf{g})$ by $x \mapsto \varphi_g(x) = \varphi(g, x)$.
The map $\varphi : G \times (\mathcal{M}, \mathbf{g}) \to (\mathcal{M}, \mathbf{g})$ is a left action if, for all $g, h \in G$, we have:
$\varphi_e = id_M$ and $\varphi_g \circ \varphi_h = \varphi_{gh}$.
Let $\varphi_h : (\mathcal{M}, \mathbf{g}) \longrightarrow (\mathcal{M}, \mathbf{g})$ denote the left group action (considered here as a multiplication) by an element $h \in G$, defined for every $x \in (\mathcal{M}, \mathbf{g})$ by: $\varphi_h(x) = h \cdot x$.

Let $\mathcal{L}_h$ denote the left regular representation of $G$ on functions $f$ defined on $\mathcal{M}$, given by $(\mathcal{L}_h f)(x) = f(\varphi_{h^{-1}}(x))$, where $h^{-1}$ is the inverse of $h \in G$.

We view a layer in a neural network as an operator. To ensure network equivariance, we require the operator to be equivariant with respect to the group actions on the corresponding function spaces.

Let $x_0$ be an arbitrary fixed point on the connected Riemannian manifold $(\mathcal{M}, \mathbf{g})$. Let $\pi : G \to (\mathcal{M}, \mathbf{g})$ denote the projection defined by associating to each element $h$ of $G$ a point in $(\mathcal{M}, \mathbf{g})$ as follows: $\forall\, h \in G$, $\pi(h) = \varphi_h(x_0)$. In other words, once a reference point $x_0 \in (\mathcal{M}, \mathbf{g})$ is chosen, the projection $\pi(h)$ associates to each element $h$ of $G$ the unique point in $(\mathcal{M}, \mathbf{g})$ to which $h$ sends $x_0$ under the action $\varphi_h$.

Let us consider a connected Lie group $G$ acting transitively on the connected Riemannian manifold $(\mathcal{M}, \mathbf{g})$. This means that for any points $x, y \in (\mathcal{M}, \mathbf{g})$, there exists an element $h \in G$ such that $\varphi_h(x) = y$, which corresponds to the definition of a homogeneous space under $G$.

**Definition 2.2** *Let $G$ be a connected Lie group with homogeneous spaces $\mathcal{M}$ and $\mathcal{N}$. Let $\phi$ be an operator mapping functions from $\mathcal{M}$ to $\mathcal{N}$. We say that $\phi$ is equivariant with respect to $G$ if, for all functions $f$, we have: $\forall\, h \in G$, $(\phi \circ \mathcal{L}_h)f = (\mathcal{L}_h \circ \phi)f$.*

**Proposition 2.1** *Let $x, y \in (\mathcal{M}, \mathbf{g})$ such that $\varphi_h(y)$ lies outside the cut locus of $\varphi_h(x)$. Then, for all $h \in G$, we have: $d_{\mathbf{g}}(x, y) = d_{\mathbf{g}}\big(\varphi_h(x), \varphi_h(y)\big)$.*

**Proof** See Appendix C.

## 3 Proposed ED2RM diffusion model

ED2RM maintains the DDPM forward process and the ELBO. The reverse process consists in leveraging PDE-GCNNs to obtain an equivariant network for noise prediction.

### 3.1 Equivariant PDEs layers-based model prediction

PDE-G-CNNs were formally introduced in homogeneous spaces with $G$-invariant tensor metric fields on quotient spaces Diop et al. (2024). Building on this foundational approach, the proposed model relies on a combination of traditional CNNs and morphological Hamilton-Jacobi PDE layers on Riemannian manifolds, and is composed of the following PDEs:

- **Convection term:**

$$\frac{\partial u}{\partial t} + \alpha u = 0 \text{ in } (\mathcal{M}, \mathbf{g}) \times (0, \infty); \quad u(\cdot, 0) = f \text{ on } (\mathcal{M}, \mathbf{g}), \tag{4}$$

where $\alpha$ is a $G$-invariant vector field on $(\mathcal{M}, \mathbf{g})$.

**Proposition 3.1** *The solution of (4) is obtained with the method of characteristics and is given by:*

$$u(x,t) = (\mathcal{L}_{h_x^{-1}} f)(\gamma_c(t)^{-1} x_0) = f(h_x \gamma_c(t)^{-1} x_0) = f(h_x \gamma_{-c}(t) x_0), \tag{5}$$

*where $h_x \in G$ satisfying $h_x x_0 = x$ for a fixed $x_0 \in M$, and $\gamma_c : \mathbb{R} \to G$ being the exponential curve such that $\gamma_c(0) = e$ and*

$$\frac{\partial}{\partial t}(\gamma_c(t)x)(t) = c(\gamma_c(t)x). \tag{6}$$

**Proof** See Smets et al. (2022).

The convection (4) is left-invariant under the action of $G$.

- **Multiscale morphological erosions and dilations:**

$$\frac{\partial u}{\partial t} \pm \|\nabla_{\mathbf{g}} u\|_{\mathbf{g}}^k = 0 \text{ in } (\mathcal{M}, \mathbf{g}) \times (0, \infty); \ u(\cdot, 0) = f \text{ on } (\mathcal{M}, \mathbf{g}), \tag{7}$$

with $k > 1$, where the positive $(+)$ sign (*resp.* negative $(-)$) stands for erosions (*resp.* dilations). The morphological operations are also equivariant with respect to $G$, ensuring the equivariance of our PDEs layers, and so for our PDEs-based network.

The output of the network is obtained as a linear combination of the outputs of each PDE layer. The above PDE system represents our stepwise PDE model solved using operator splitting, where each step corresponds to one of the above PDEs.

The connection between multi-scale morphological dilations and erosions had already been established by solving a first-order Hamilton–Jacobi type PDE in $\mathbb{R}^n$. Their extensions in Riemannian manifolds can be provided by properly defining the related Hamiltonian.

Let $(\mathcal{M}, \mathbf{g})$ be a compact, connected Riemannian manifold equipped with a metric $\mathbf{g}$, and let $f, b : (M, \mathbf{g}) \longrightarrow \mathbb{R}$. Let $T\mathcal{M}$ denote the tangent bundle of $(\mathcal{M}, \mathbf{g})$, and let $L : T\mathcal{M} \to \mathbb{R}$ be a Lagrangian function. Let $T^*\mathcal{M}$ denote the cotangent bundle of $(\mathcal{M}, \mathbf{g})$, and let us define the Hamiltonian $H : T^*\mathcal{M} \to \mathbb{R}$ associated with the Lagrangian $L$ by:

$$H(x, q) = \sup_{v \in T_x \mathcal{M}} \{q(v) - L(x, v)\}.$$

The Hamilton–Jacobi PDE can be extended to Riemannian manifolds as follows:

$$\frac{\partial u}{\partial t} + H(x, \nabla u) = 0 \quad \text{in } (\mathcal{M}, \mathbf{g}) \times (0, +\infty); \quad u(\cdot, 0) = f \text{ on } (\mathcal{M}, \mathbf{g}). \tag{8}$$

PDE (8) admits viscosity solutions Fathi (2008); Diop et al. (2021). Multi-scale morphological erosions (*resp.* dilations) are obtained by setting $H = \|\nabla_{\mathbf{g}} u\|_{\mathbf{g}}^k$ (*resp.* $H = -\|\nabla_{\mathbf{g}} u\|_{\mathbf{g}}^k$) in equation 8.

**Proposition 3.2** *Let $k > 1$ and let $f \in C^0((\mathcal{M}, \mathbf{g}), \mathbb{R})$ be a continuous function. The viscosity solutions to the Cauchy problem:*

$$\frac{\partial u}{\partial t} + \|\nabla_{\mathbf{g}} u\|_{\mathbf{g}}^k = 0 \quad \text{in } (\mathcal{M}, \mathbf{g}) \times (0, +\infty); \quad u(\cdot, 0) = f \text{ on } (\mathcal{M}, \mathbf{g}), \tag{9}$$

*are given by:*

$$u(t, x) = \inf_{h \in G} \left\{ f(\varphi_h(x_0)) + c_k \frac{d_{\mathbf{g}}(\varphi_{h^{-1}}(x), x_0)^{\frac{k}{k-1}}}{t^{\frac{1}{k-1}}} \right\}, \text{ where } c_k = \frac{k-1}{k^{\frac{k}{k-1}}}. \tag{10}$$

**Proof** See Diop et al. (2024).

Morphological multi-scale Riemannian dilations at scale $t$ are obtained by reversing the time:

$$\frac{\partial w}{\partial t} - \|\nabla_{\mathbf{g}} w\|_{\mathbf{g}}^k = 0 \quad \text{in } (\mathcal{M}, \mathbf{g}) \times (0, +\infty); \quad w(\cdot, 0) = f \text{ on } (\mathcal{M}, \mathbf{g}). \tag{11}$$

The viscosity solutions of the Cauchy problem are obtained in a similar way as:

$$w(t, x) = \sup_{h \in G} \left\{ f\big(\varphi_h(x_0)\big) - c_k \frac{d_{\mathbf{g}}\big(\varphi_{h^{-1}}(x), x_0\big)^{\frac{k}{k-1}}}{t^{\frac{1}{k-1}}} \right\}. \tag{12}$$

Let us consider the family of functions $(b_t^k)$ defined by: $b_t^k = c_k \dfrac{d_\mu(x_0, \cdot)^{\frac{k}{k-1}}}{t^{\frac{1}{k-1}}}$. The case $k = 2$ corresponds to quadratic structuring functions. Letting $k > 1$ allows us to deal with more general structuring functions than quadratic ones, leading to a better handling of thin image (data) structures. Let us introduce the notion of group Riemannian morphological convolution as follows:

**Definition 3.1** *The group morphological convolution $\Diamond$ between $b$ and $f$ is defined for all $x \in (\mathcal{M}, \mathbf{g})$ by:*

$$b \Diamond f(x) = \inf_{p \in G} \{ f(\varphi_p(x_0)) + b(\varphi_{p^{-1}}(x)) \}.$$

Thanks to Definition 3.1, morphological multi-scale Riemannian erosions (10) and dilations (12) at scale $t$ can respectively write:

$$u(t, x) = b_t^k \Diamond f(x) \text{ and } w(t, x) = -(b_t^k \Diamond (-f))(x). \tag{13}$$

### 3.2 EXAMPLE OF COMPACT RIEMANNIAN MANIFOLD $\mathcal{M}$: HYPERBOLIC BALL

We choose, as an example of a Riemannian manifold $\mathcal{M}$, the hyperbolic ball and use it for our numerical experiments, as it provides a natural framework to represent equivariance with respect to the group $E(n)$. Its negatively curved geometry allows for effective capture of hierarchical and non-local relationships between points, while ensuring that distances, invariant under the transformations of the group $E(n)$ (reflections, rotations, and permutations), are preserved. This facilitates the definition of stable equivariant operators and improves learning in neural networks based on PDE-GCNNs.

Let us consider the hyperbolic ball defined by:

$$\mathbb{B}^n = \left\{ (x_1, \ldots, x_n) \in \mathbb{R}^n \mid \sum_{i=0}^n x_i^2 < 1 \right\}. \tag{14}$$

In next steps, we endow $\mathbb{B}^n$ with a metric $g$, and we show that the hyperbolic distance $d_{\mathbb{B}^n}$ is invariant under translations, rotations, reflections, and permutations. We also show an embedding of $\mathbb{R}^n$ into $\mathbb{B}^n$, which will preserve data (here image) structures within the hyperbolic ball $\mathbb{B}^n$, enabling the desired equivariance. For the numerical computations, we take $\mathcal{M} = \mathbb{B}^2$.

Let us consider the following metric $g$ in $\mathbb{B}^n$:

$$\mathbf{g} = \frac{4(dx_1^2 + \ldots + dx_n^2)}{(1 - \|x\|^2)^2}, \tag{15}$$

where $\|\cdot\|$ represents the Euclidean norm in $\mathbb{R}^n$. The length of a curve $\gamma : [a, b] \to \mathbb{B}^n$ is given by:

$$L(\gamma) = \int_a^b \sqrt{g(\gamma'(t), \gamma'(t))} \mathrm{d}t = \int_a^b 2 \frac{\sqrt{\gamma_1'(t)^2 + \ldots + \gamma_n'(t)^2}}{\sqrt{1 - (\gamma_1(t)^2 + \ldots + \gamma_n(t)^2)}} \mathrm{d}t, \tag{16}$$

where $\gamma(t) = (\gamma_1(t), \ldots, \gamma_n(t))$. The distance between two points $x, y \in \mathbb{B}^n$ is the infimum over all curves that join $x$ and $y$. Then, the hyperbolic distance $d_{\mathbb{B}^n}(x, y)$ between $x$ and $y$ is given by:

$$\cosh d_{\mathbb{B}^n}(x, y) = 1 + \frac{2\|x - y\|^2}{(1 - \|x\|^2)(1 - \|y\|^2)}, \tag{17}$$

and thus, we derive:

$$\mathrm{d}_{\mathbb{B}^n}(x, y) = \mathrm{Argcosh}\left(1 + \frac{2\|x - y\|^2}{(1 - \|x\|^2)(1 - \|y\|^2)}\right).\tag{18}$$

The group $E(n)$ consists of translations, rotations, reflections, and permutations in $\mathbb{R}^n$. Since $\mathrm{d}_{\mathbb{B}^n}$ depends only on the Euclidean norm, which is invariant under Euclidean isometries, $\mathrm{d}_{\mathbb{B}^n}$ is invariant under all elements of $E(n)$.

**Proposition 3.3** $\mathrm{d}_{\mathbb{B}^n}$ *is invariant under Euclidean transformations.*

**Proof** See Appendix D.

Let $\Phi$ be the mapping defined from $\mathbb{R}^n$ to $\mathbb{B}^n$ by:

$$\Phi: \quad \mathbb{R}^n \longrightarrow \mathbb{B}^n; \; x \mapsto \frac{x}{\sqrt{1 + \|x\|^2}},\tag{19}$$

where $\|\cdot\|$ denotes the Euclidean norm in $\mathbb{R}^n$. $\Phi$ is well-defined because, $\forall x \in \mathbb{R}^n$, we have:

$$\|\Phi(x)\|^2 = \frac{\|x\|^2}{1 + \|x\|^2} < 1.\tag{20}$$

**Proposition 3.4** $\Phi$ *is an embedding of $\mathbb{R}^n$ into $\mathbb{B}^n$.*

**Proof** See Appendix E.

### 3.3 PDEUnet Network Architecture

Our architecture (Fig. 1(a)) is based on a classical U-Net structure, as used for noise prediction in DDPM Ho et al. (2020), but we integrate our previously defined PDE layer in order to obtain a new network that is equivariant with respect to the group $E(n)$.

A diffusion U-Net typically consists of three main components: an encoder (*Downsampling*), a decoder (*Upsampling*), and a middle block (*Middle Block*). The latter, located between the encoder and the decoder, corresponds to the lowest spatial resolution and the highest number of channels. It plays a crucial role in merging, transforming, and refining the representations extracted by the encoder before their reconstruction by the decoder. It often consists of residual blocks for deeper feature extraction and may include an attention module to capture long-range dependencies. In ED2RM, we propose a novel modification in the middle block. Specifically, we replace the classical ResNetBlocks, commonly used in standard U-Nets, with *ResnetCDEBlocks*. As illustrated in Fig. 1(b), these blocks enhance predictive capacity while introducing explicit equivariance with respect to the group $E(n)$.

To the best of our knowledge, such a combination of a diffusion U-Net with residual blocks of the CDE type (Convection, Dilation, and Erosion) has not yet been reported in the literature. This architecture therefore represents an original contribution aimed at strengthening both the robustness and expressiveness of diffusion models.

## 4 Experiments

To evaluate the performance of our diffusion model on the MNIST and Rotated MNIST datasets, we trained the model with a batch size of 64 images over a total of 60,000 iterations. Optimization was performed using the Adam algorithm ($\beta_1 = 0.9, \beta_2 = 0.99$) with a learning rate of $1 \times 10^{-4}$. An Exponential Moving Average (EMA) update was applied every 10 iterations with a decay factor of 0.995 to stabilize training and improve the quality of the generated images.

Test image generation was conducted every 100 samples, with each sample producing 25 images, in order to monitor both the visual and quantitative evolution of quality. FID scores were computed at each step over 2,500 images to assess the fidelity and diversity of the generated samples. We also employed additional metrics to further evaluate both models.

In this study, we chose to compare our approach only with the baseline DDPM model. This choice is motivated by two main reasons. First, DDPM is the foundational model of diffusion-based approaches and serves as a reference in the majority of works in the literature. It therefore provides a relevant baseline for objectively evaluating the improvements introduced by our method. Second, our objective is not to outperform all existing variants of diffusion models, but rather to quantify the specific impact of incorporating equivariance and PDE-G-CNNs on the performance of the original model. By restricting our comparison to this baseline, we are able to more rigorously isolate and analyze the actual contribution of our approach.

In the following, we present the results in the form of tables and graphs, along with a set of generated image samples from each dataset and using each model.

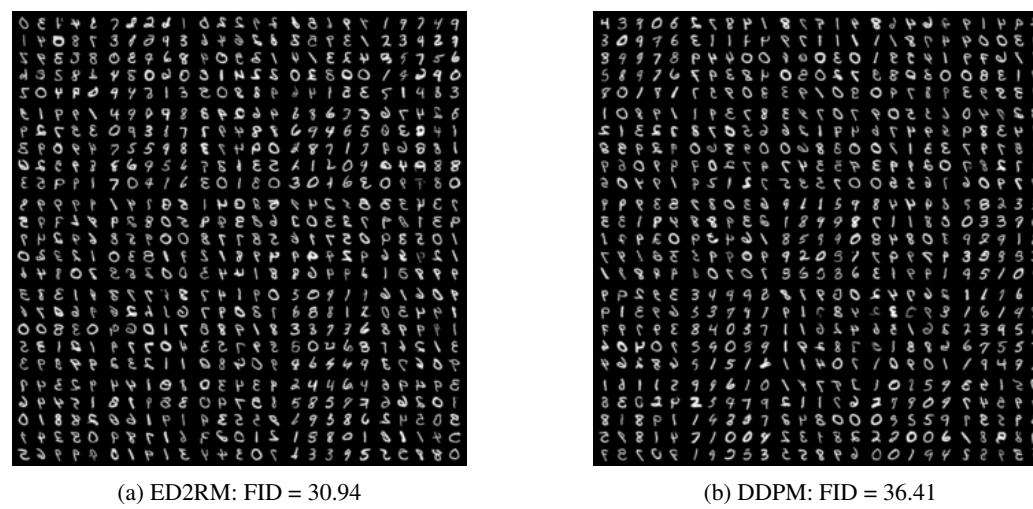

(a) ED2RM: FID = 30.94          (b) DDPM: FID = 36.41

Figure 2: Generated samples using ED2RM vs. DDPM on MNIST.

Fig. 2 shows the samples with the best FID scores during training using ED2RM (Fig. 2a) and DDPM (Fig. 2b). Visual inspection indicates that the quality of the generated samples is generally similar, with respective FID scores of 30.94 and 36.41. These results suggest that the ED2RM model achieves slightly better generation performance on the MNIST dataset.

We report on Table 1 the quantitative results obtained using different metrics, namely FID and mean IS of the generated samples. Obtained ED2RM and DDPM Results show comparable performance in terms of quality and diversity on MNIST. We can notice that ED2RM achieves a slightly lower mean FID (45.14) compared to DDPM (46.91). In addition, Fig. 4a illustrates the FID evolution during training, and it highlights that ED2RM produces higher-quality samples during the first thirty training steps. The DDPM model subsequently reaches a comparable level of quality after the thirtieth step. This indicates that ED2RM converges faster towards generating high-quality samples, while DDPM requires more iterations to reach similar performance.

| Metric | MNIST | | ROTOMNIST | |
|---|---|---|---|---|
| | ED2RM | DDPM | ED2RM | DDPM |
| FID | 45.14 | 46.91 | 49,30 | 173,44 |
| IS | $1.207 \pm 0.242$ | $1.207 \pm 0.242$ | $1.327 \pm 0.175$ | $1.298 \pm 0.139$ |

Table 1: ED2RM vs. DDPM on MNIST and ROTOMNIST.

On the RotoMNIST dataset, as shown in Fig. 3, the samples generated using ED2RM (Fig. 3a) exhibit significantly higher quality, with an FID score of 35.75, whereas DDPM (Fig. 3b) presents a much higher FID of 150.17. This discrepancy is further confirmed in Table 1, which shows that ED2RM achieves superior FID and IS values, demonstrating its ability to generate samples that are both higher in quality and more diverse than those produced by DDPM.

Moreover, Fig. 4b, illustrating the FID evolution throughout the training steps on the RotoMNIST dataset, highlights a substantial performance gap between the two models over the course of training. It can be observed that ED2RM effectively adapts to the RotoMNIST data and maintains a

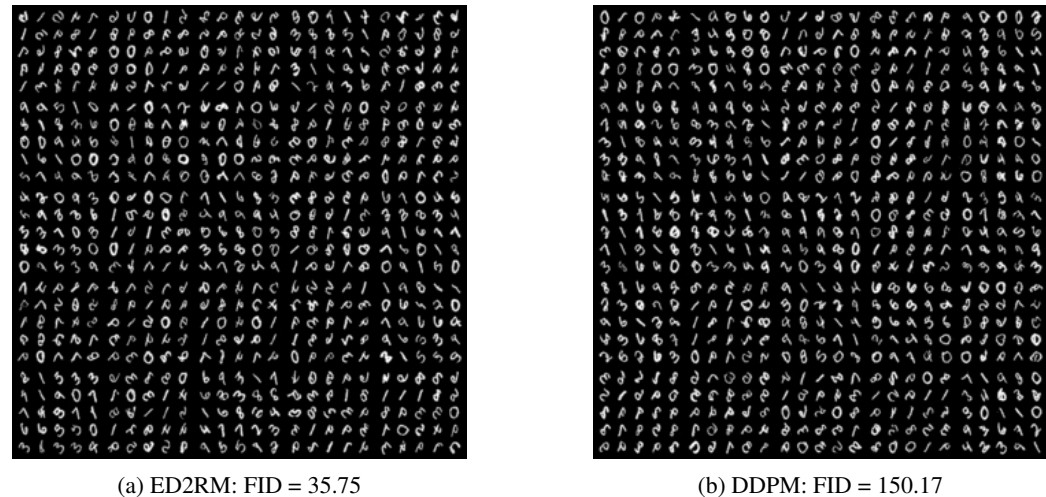

(a) ED2RM: FID = 35.75           (b) DDPM: FID = 150.17

Figure 3: Generated samples using ED2RM vs. DDPM on RotoMNIST.

performance level comparable to that achieved on the MNIST dataset. In contrast, DDPM fails to sustain similar performance with an equivalent number of training steps.

In conclusion, both models achieve similar results on the MNIST dataset, although DDPM requires more training steps to reach performance levels comparable to ED2RM. However, on the RotoM-NIST dataset, DDPM does not maintain the performance observed on MNIST, while ED2RM preserves nearly the same generation quality across both datasets. This stability can be attributed to ED2RM's inherent equivariance to group transformations.

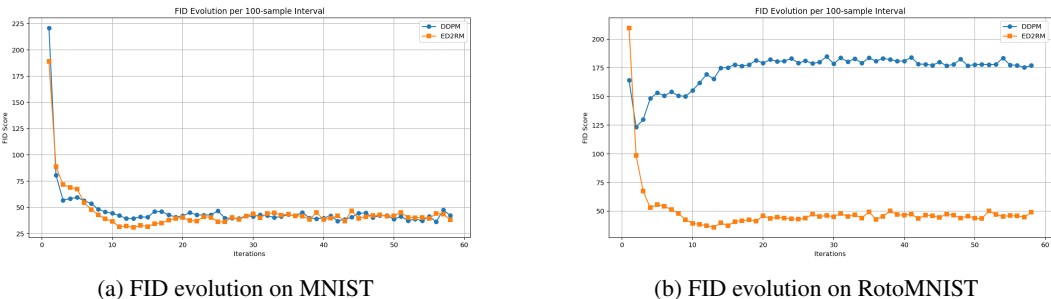

(a) FID evolution on MNIST           (b) FID evolution on RotoMNIST

Figure 4: FID evolution during training using ED2RM vs. DDPM on MNIST and RotoMNIST.

## 5   CONCLUSION

We have proposed here an equivariant denoising diffusion model (ED2RM) that integrates morphological PDEs within group-equivariant CNNs on Riemannian manifolds. By leveraging convection and Hamilton–Jacobi PDEs, the framework not only preserves key data symmetries—translations, rotations, reflections, and permutations—but also enhances geometric feature extraction in the denoising process. Our experiments on MNIST and RotoMNIST confirmed that ED2RM achieves faster convergence, superior FID scores, and improved robustness compared to standard DDPM, particularly under geometric transformations. These results highlight the potential of combining PDE-based equivariant architectures with diffusion models to produce more interpretable and resilient generative frameworks. Future work will explore real image synthetis and scaling ED2RM to higher-dimensional datasets and extending its applicability to 3D shape generation and molecular modeling.

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

## A  DETAILS ON PROBABILISTIC DIFFUSION MODELS

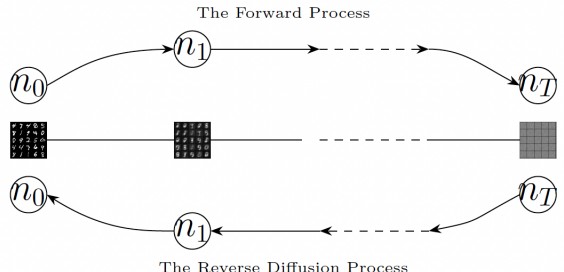

Figure 5: Illustration of the forward noising process and the inverse denoising process.

Given observations $x \sim q(x)$, the model operates on latent variables $n_0, \ldots, n_T$ of the same dimension as $x$, where $n_0$ corresponds to the observation $x$ and $n_T$ represents standard Gaussian noise (see Fig. 5).

**Forward Process.** The forward process is Markovian; thus, for all $t \in \{0, \ldots, T\}$, $n_t$ depends only on $n_{t-1}$ and not on earlier variables **?**:

$$q(n_t \mid n_{t-1}, n_{t-2}, \ldots, n_0) = q(n_t \mid n_{t-1}) \tag{21}$$

Hence, the joint distribution of this process can be written as:

$$q(n_1, n_2, \ldots, n_T \mid n_0) = \prod_{t=1}^{T} q(n_t \mid n_{t-1}) \tag{22}$$

For any $t > s$, the transition distribution from step $s$ to $t$ can be defined using the Gaussian reparameterization of **?**, considering a standard Gaussian $\varepsilon \sim \mathcal{N}(0, I)$. Thus, Equation (1) can be rewritten as:

$$n_t = \sqrt{\alpha_t} n_{t-1} + \sqrt{1 - \alpha_t}\, \varepsilon \tag{23}$$

Consequently, for any $t > s$, we have:

$$q(n_t \mid n_s) = \mathcal{N}(n_t : \sqrt{\alpha_{t/s}} n_s, \Gamma_{t/s} I) \tag{24}$$

with $\alpha_{t/s} = \prod_{i=s+1}^{t} \alpha_i$ and $\Gamma_{t/s} = 1 - \alpha_{t/s}$. Relative to the initial data $n_0$:

$$q(n_t \mid n_0) = \mathcal{N}(n_t : \sqrt{\bar{\alpha}} n_0, \bar{\Gamma} I) \tag{25}$$

where $\bar{\alpha} = \prod_{i=1}^{t} \alpha_i$ and $\bar{\Gamma} = 1 - \bar{\alpha}$.

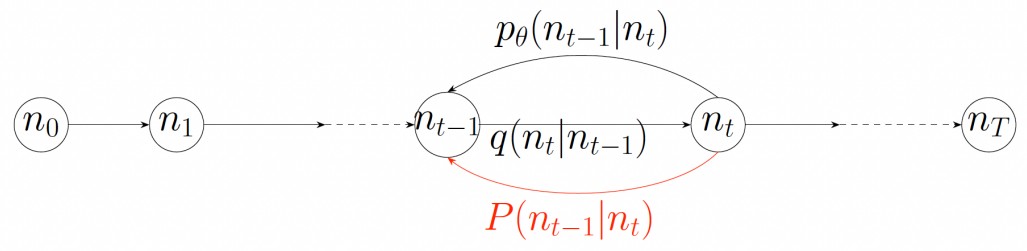

Figure 6: Conditional distributions in the forward and generative processes of the diffusion model.

**Inverse Generative Process.** The generative process is also modeled as a first-order Markov chain, i.e.,

$$P(n_{t-1} \mid n_t, n_{t+1}, \ldots, n_T) = P(n_{t-1} \mid n_t) \tag{26}$$

Thus, the joint distribution of the generative process can be written as:

$$p_\theta(n_{0:T}) = P(n_T) \prod_{t=1}^{T} p_\theta(n_{t-1} \mid n_t) \tag{27}$$

with $P(n_T)$ typically defined as standard Gaussian noise:

$$P(n_T) = \mathcal{N}(0, I) \tag{28}$$

The true distribution is similarly defined as in Equation (eq:eq10):

$$P(n_{t-1} \mid n_t, n_0) = \mathcal{N}(n_{t-1} : \tilde{\mu}(n_t, n_0), \tilde{\sigma}^2 I) \tag{29}$$

Using the same reparameterization technique as in the forward case (see Equation (23)), we can sample from a standard Gaussian $\varepsilon \sim \mathcal{N}(0, I)$. Then, the mean in Equation (29) can be expressed as:

$$\tilde{\mu}(n_t, n_0) = \frac{1}{\sqrt{\alpha_t}} \left( n_t - \frac{\Gamma_t}{\sqrt{1 - \bar{\alpha}_t}} \varepsilon \right) \tag{30}$$

Similarly, an expression for $\mu_\theta(n_t, t)$ in Equation (2) is given. Since the learned denoising process is defined from the true denoising process, we have:

$$\mu_\theta(n_t, t) = \frac{1}{\sqrt{\alpha_t}} \left( n_t - \frac{\Gamma_t}{\sqrt{1 - \bar{\alpha}_t}} \varepsilon_\theta(n_t, t) \right) \tag{31}$$

where $\varepsilon_\theta(n_t, t) = \phi(n_t, t)$ is the output of the neural network $\phi$ at iteration $t$.

**Variational Lower Bound of the Likelihood.** As mentioned earlier, diffusion models introduce a sequence of latent variables. The data likelihood is written as:

$$p_\theta(n_0) = \int p_\theta(n_0, n_{1:T}) \, dn_{1:T}.$$

Direct maximization of this likelihood is intractable; therefore, diffusion models optimize a variational lower bound (ELBO) on the data likelihood:

$$\mathcal{L} := \mathbb{E}_{q(n_{1:T}|n_0)}[\log p_\theta(n_0 \mid n_{1:T})] - \mathrm{KL}(q(n_{1:T} \mid n_0) \, \| \, p_\theta(n_{1:T})) \; \leq \; \log p_\theta(n_0), \tag{32}$$

where $\mathbb{E}[\cdot]$ denotes expectation and $\mathrm{KL}(\cdot\|\cdot)$ is the Kullback–Leibler divergence.

Expanding, we obtain an equivalent expression (see Ho et al. (2020); Sohl-Dickstein et al. (2015)):

$$\mathcal{L}(\theta) = -\mathrm{KL}(q(n_T \mid n_0) \, \| \, p(n_T)) - \sum_{t=2}^{T} \mathrm{KL}(q(n_{t-1} \mid n_t, n_0) \, \| \, p_\theta(n_{t-1} \mid n_t))$$
$$+ \mathbb{E}_{q(n_{1:T}|n_0)}[\log p_\theta(n_0 \mid n_1)].$$

The ELBO must be maximized with respect to $\theta$. The first KL divergence is independent of $\theta$ and can therefore be ignored during optimization. Hence, maximizing $\mathcal{L}$ reduces to:

$$\underset{\theta}{\text{minimize}} \ \sum_{t=2}^{T} \text{KL}(q\,(n_{t-1} \mid n_t, n_0) \,\|\, p_\theta\,(n_{t-1} \mid n_t)) - \mathbb{E}_{q(n_{1:T}|n_0)}[\log p_\theta\,(n_0 \mid n_1)]. \quad (33)$$

This equation trains the inverse distribution $p_\theta(n_{t-1} \mid n_t)$ to match the true denoising distribution $q(n_{t-1} \mid n_t, n_0)$ by minimizing their KL divergence. It can thus be used as a loss function for a neural network parameterized by $\theta$, emphasizing the alignment between these two distributions.

In other words, optimizing the ELBO forces the model to learn a denoising process capable of reversing the progressive diffusion of noise. Training consists of bringing the learned inverse process $p_\theta$ closer to the true denoising process $q$, while maximizing the likelihood of the observed data.

## B  BACKGROUND ON MORPHOLOGICAL OPERATORS AND PDEs

Let $b : \mathbb{R}^2 \to \bar{\mathbb{R}}$ be a concave function, known also as the structuring function or convolution kernel. Let us consider the subset $\mathbb{E}$ of $\mathbb{Z}^2$ and the function $f : \mathbb{E} \to \bar{\mathbb{R}}$.

**Definition B.1** *Morphological dilation and erosion are respectively defined as:*

$$f \oplus b(x) = \sup_{y \in \mathbb{E}}[f(y) + b(x - y)] \quad (34)$$

$$f \ominus b(x) = \inf_{y \in \mathbb{E}}[f(y) - b(y - x)]. \quad (35)$$

Let $B \subseteq \mathbb{E}$ be a bounded set. A flat structuring function (SF) satisfies $b(x) = 0$ if $x \in B$ and $b(x) = -\infty$ if $x \in B^c$. The flat morphological dilation and erosion respectively write:

$$f \oplus B(x) = \sup_{y \in B}[f(x - y)] \text{ and } f \ominus B(x) = \inf_{y \in B}[f(x + y)]. \quad (36)$$

As for an interpretation, erosion shrinks positive peaks, and peaks thinner than the structuring function disappear. One has the dual effects for morphological flat dilation. Both the morphological dilation and erosion are translation invariant.

**Definition B.2** *Let $\mathcal{F}$ be a family of real functions defined on $\Omega \subseteq \mathbb{R}^2$. We say that $T : \mathcal{F} \to \mathcal{F}$ is said to be increasing (monotone) if and only if it satisfies:*
$\forall f_1, f_2 \in \mathcal{F}$ *such that* $(f_1 \geq f_2$ *on* $\Omega)$ *implies* $(T(f_1) \geq T(f_2)$ *on* $\Omega)$.

**Proposition B.1** *Morphological dilation and erosion satisfy the following duality and adjunction properties:*

1. *duality:* $f \oplus b = -(-f \ominus b)$

2. *adjunction:* $(f_1 \oplus b \leq f_2$ *on* $E) \iff (f_1 \leq f_2 \ominus b$ *on* $E)$.

Let $(b_t)_{t \geq 0}$ the family of structuring functions defined by using the SF $b$, as follows:

$$b_t(x) = \begin{cases} tb(x/t) & \text{for } t > 0 \\ 0 & \text{for } t = 0, \ x = 0 \\ -\infty & \text{otherwise.} \end{cases}$$

The family $(b_t)_{t \geq 0}$ satisfies the semi-group property:
$\forall \, s, t \geq 0, \ (b_s \,\bar{\oplus}\, b_t)(x) = b_{s+t}(x, y)$.

**Definition B.3** *Morphological multiscale dilations and erosions are defined as follows:*

$$(f \oplus b_t)(x) = \sup_{y \in \mathbb{E}}[f(y) + b_t(x - y)] \quad (37)$$

$$(f \ominus b_t)(x) = \inf_{y \in \mathbb{E}}[f(y) - b_t(y - x)]. \quad (38)$$

Considering flat structuring function (SF), morphological multiscale dilations and erosions are obtained equivalently by considering $B_t = tB$ as multiscale SFs.

The link between morphological scale-spaces and PDEs was established by running the following PDE that performs multiscale flat dilations and erosions on a given image $f$ Meyer & Maragos (2000); Schmidt & Weickert (2016):

$$\partial_t u \pm \|\nabla u\| = 0; \ u(\cdot, 0) = f. \tag{39}$$

Depending on the shape of SF, different PDEs can be obtained. For instance, considering the sets $S_p = \left\{ x = (x_1, x_2) \in \mathbb{R}^2 : |x|_p \leq 1 \right\}$, where $|\cdot|_p$ is the $L^p$ norm, one gets:

- for a square $S_1$: $\partial_t u \pm \|\nabla u\|_1 = 0; \ u(\cdot, 0) = f$
- for a dis $S_2$: $\partial_t u \pm \|\nabla u\|_2 = 0; \ u(\cdot, 0) = f$
- for a rhombus $S_\infty$: $\partial_t u \pm \|\nabla u\|_\infty = 0; \ u(\cdot, 0) = f$.

Notice that PDE (39) is a special case of first order Hamilton-Jacobi equation type, which can be formulated in a more general form as follows:

$$\begin{cases} \dfrac{\partial u(x,t)}{\partial t} + H\left(x, \nabla u(x,t)\right) = 0 \text{ on } \mathbb{R}^n \times (0, +\infty) \\ u(\cdot, 0) = f \text{ on } \mathbb{R}^n. \end{cases} \tag{40}$$

General Hamilton-Jacobi equation is studied in a viscosity sense, because there is no classical solution for such equations. For a convex Hamiltonian $H$ and some regularity on $f$, the viscosity solution is given by Hopf-Lax formulas Barron (2021); Donato (2023):

$$u(x,t) = \inf_{y \in \mathbb{R}^n} \left\{ f(y) + tL\left(\frac{x-y}{t}\right) \right\}, \tag{41}$$

where $L$ is the Lagrangian, defined as the Legendre-Fenchel transform of $H$.

## C  PROOF OF PROPOSITION 2.1

For all $x \in (M, g)$, we refer to $G$-invariance of vector fields $X : x \mapsto T_x M$ if $\forall h \in G$ and for all differentiable functions $f$, one has:

$$X(x)f = X(\varphi_h(x))[\mathcal{L}_h f]. \tag{42}$$

**Definition C.1** *A vector field $X$ on $(M, g)$ is invariant with respect to $G$ if $\forall h \in G$ and $\forall x \in (M, g)$, one has:*

$$X(\varphi_h(x)) = (\varphi_h)_* X(x). \tag{43}$$

**Definition C.2** *A $(0, 2)$-tensor field $g$ on $M$ is $G$-invariant if $\forall h \in G$, $\forall x \in M$ and $\forall v, w \in T_x(M)$, one has:*

$$g|_h(v, w) = g|_{\varphi_h(x)}((\varphi_h)_* v, (\varphi_x)_* w). \tag{44}$$

It follows from Definition C.2 that properties derived from metric tensor field $G$ invariance and vector field $G$ invariance are the same.

**Definition C.3** *Let $(M, g)$ a connected Riemannian manifold, $x, y \in (M, g)$. The distance between $x$ and $y$ is defined as follows:*

$$\mathrm{d}_g(x, y) = \inf_{\gamma \in \Gamma_t(x,y)} \int_0^t \sqrt{g|_{\gamma(t)}(\dot{\gamma}(t), \dot{\gamma}(t))} \mathrm{d}t, \tag{45}$$

*with $\Gamma_t(x, y) = \{\gamma : [0, t] \longrightarrow (M, g) \text{ of class } C^1, \gamma(0) = x \text{ and } \gamma(t) = y\}$.*

**Definition C.4** *The cut locus is defined as the set of points $x \in M$ (or $h \in G$) from which the distance map is not smooth (except at $x$ or $h$).*

**Proof** Let us perform a left multiplication by $h$ in one direction and by $h^{-1}$ in the other direction. A bijection can then be established between $C^1$ curves connecting $x$ to $y$ and connecting $\varphi_h(x)$ to $\varphi_h(y)$. Thus, we have:

$$d_g\big(\varphi_h(x), \varphi_h(y)\big) = \inf_{\beta \,\in\, \Gamma_t(\varphi_h(x), \varphi_h(y))} \int_0^t \sqrt{g|_{\beta(t)}(\dot{\beta}(t), \dot{\beta}(t))}\,\mathrm{d}t,$$

$$= \inf_{h\gamma \,\in\, \Gamma_t(\varphi_h(x), \varphi_h(y))} \int_0^t \sqrt{g|_{h\gamma(t)}\Big(\varphi_h(\dot{\gamma}(t)), \varphi_h(\dot{\gamma}(t))\Big)}\,\mathrm{d}t \quad \text{with } \gamma \in \Gamma_t(\varphi_h(x), \varphi_h(y))$$

$$= \inf_{h\gamma \,\in\, \Gamma_t(\varphi_h(x), \varphi_h(y))} \int_0^t \sqrt{g|_{h\gamma(t)}\Big((\varphi_h)_*\dot{\gamma}(t), (\varphi_h)_*\dot{\gamma}(t)\Big)}\,\mathrm{d}t$$

$$= \inf_{h\gamma \,\in\, \Gamma_t(\varphi_h(x), \varphi_h(y))} \int_0^t \sqrt{g|_{\gamma(t)}(\dot{\gamma}(t), \dot{\gamma}(t))}\,\mathrm{d}t \qquad \text{by equation 44}$$

$$= \inf_{\gamma \,\in\, \Gamma_t(x,y)} \int_0^t \sqrt{g|_{\gamma(t)}\big(\dot{\gamma}(t), \dot{\gamma}(t)\big)}\,\mathrm{d}t = d_g(x,y) \quad \blacksquare$$

## D   PROOF OF PROPOSITION 3.3

**Proof** The case $n = 2$ is trivial. Let us prove the result for $n = 3$; the general case follows the same way.

### • Rotations and Reflections

Let $\{\vec{u}, \vec{v}, \vec{w}\}$ be an orthonormal basis of $\mathbb{R}^3$. Define $R_\theta$ in this basis as:

$$R = \begin{pmatrix} -1 & 0 & 0 \\ 0 & \cos\theta & -\sin\theta \\ 0 & \sin\theta & \cos\theta \end{pmatrix}, \tag{46}$$

which represents an anti-rotation by angle $\theta$ around $\vec{u}$ (a composition of rotation and reflection). Applying $R_\theta$ to $X = (x, y, z)$ yields:

$$R_\theta X = \begin{pmatrix} -x \\ y\cos\theta - z\sin\theta \\ y\sin\theta + z\cos\theta \end{pmatrix}. \tag{47}$$

Its Euclidean norm satisfies:

$$\|R_\theta X\|^2 = \|X\|^2, \tag{48}$$

and similarly $\|R_\theta X - R_\theta Y\| = \|X - Y\|$. Substituting these into the expression for $d_{\mathbb{B}^3}$, we obtain:

$$d_{\mathbb{B}^3}(R_\theta X, R_\theta Y) = d_{\mathbb{B}^3}(X, Y). \tag{49}$$

### • Permutations

Let us represent the group of permutations of $\{1, 2, 3\}$ as follows:

$$\sigma = \begin{pmatrix} 1 & 2 & 3 \\ \sigma(1) & \sigma(2) & \sigma(3) \end{pmatrix}, \tag{50}$$

with $\sigma(1) = 2, \sigma(2) = 3, \sigma(3) = 1$. It follows that $\|\sigma X\| = \|X\|$ and $\|\sigma X - \sigma Y\| = \|X - Y\|$, hence:

$$d_{\mathbb{B}^3}(\sigma X, \sigma Y) = d_{\mathbb{B}^3}(X, Y). \quad \blacksquare \tag{51}$$

## E   PROOF OF PROPOSITION 3.4

**Proof** $\Phi$ is well-defined and continuous. Next, we show that $\Phi$ is an injection and a $\mathcal{C}^k$-diffeomorphism.

- **Injectivity of $P$**

Let $x, y \in \mathbb{R}^n$. We assume that $\Phi(x) = \Phi(y)$, we need to show that $x = y$.

$$\Phi(x) = \Phi(y) \implies \frac{x}{\sqrt{1 + \|x\|^2}} = \frac{y}{\sqrt{1 + \|y\|^2}}$$
$$\implies \frac{\|x\|}{\sqrt{1 + \|x\|^2}} = \frac{\|y\|}{\sqrt{1 + \|y\|^2}}$$
$$\implies \|x\|^2 = \|y\|^2$$
$$\implies x = y.$$

Hence, $\Phi$ is injective.

- **$\mathcal{C}^k$-diffeomorphism property of $\Phi$:**

For $x = (x_i)_{i=1}^n \in \mathbb{R}^n$, we have

$$\Phi(x) = \frac{x}{\sqrt{1 + \|x\|^2}} = \left( \frac{x_i}{\sqrt{1 + \|x\|^2}} \right)_{i=1}^n, \tag{52}$$

and we denote $\Phi_i(x) = \dfrac{x_i}{\sqrt{1 + \|x\|^2}}$. Then:

$$\frac{\partial}{\partial x_j} \Phi_i(x) = \frac{\delta_{ij}}{\sqrt{1 + \|x\|^2}} - \frac{x_i x_j}{(1 + \|x\|^2)^{3/2}},$$

where

$$\delta_{ij} = \begin{cases} 1 & \text{if } i = j, \\ 0 & \text{otherwise,} \end{cases} \tag{53}$$

is the Kronecker symbol. Thus, the Jacobian matrix of $\Phi$ is:

$$J_{\Phi(x)} = \frac{1}{\sqrt{1 + \|x\|^2}} \left( I - \frac{x \otimes x}{1 + \|x\|^2} \right),$$

where $\otimes$ denotes the tensor product. In $\mathbb{R}^3$, we obtain:

$$\det(J_{\Phi(x)}) = 1 - \frac{\|x\|^2}{\sqrt{1 + \|x\|^2}} \neq 0 \quad \forall x \in \mathbb{R}^3,$$

which shows that $J_{\Phi(x)}$ is invertible. Hence, $\Phi$ is a diffeomorphism onto its image, and therefore, it is an embedding of $\mathbb{R}^n$ into $\mathbb{B}^n$. ∎

# F  ADDITIONAL QUALITATIVE RESULTS

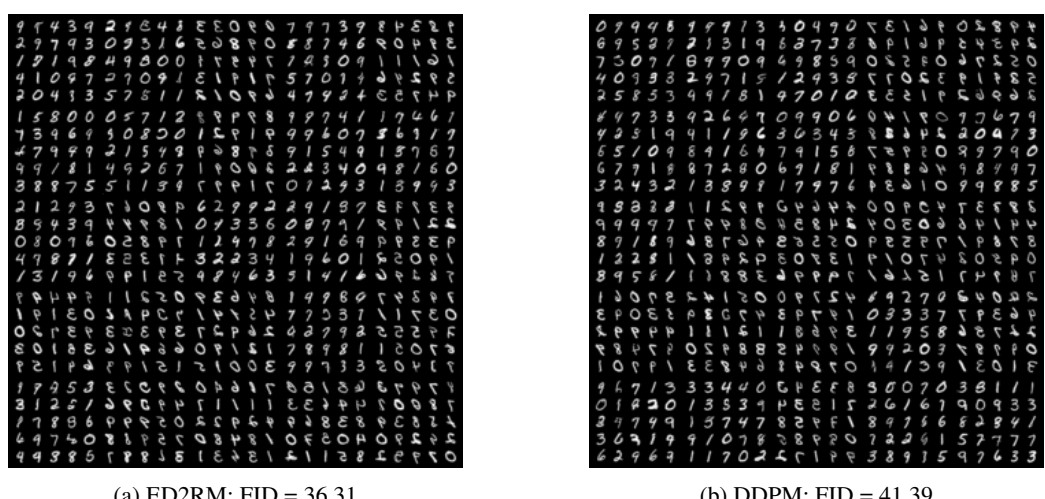

(a) ED2RM: FID = 36.31

(b) DDPM: FID = 41.39

Figure 7: Generated image samples from our ED2RM model and the DDPM baseline on the MNIST dataset (Sample 30).

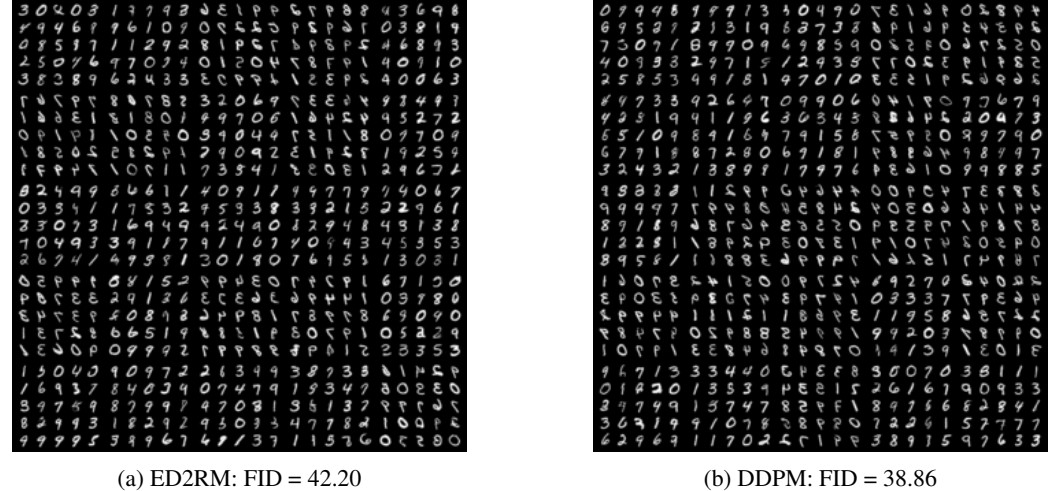

(a) ED2RM: FID = 42.20

(b) DDPM: FID = 38.86

Figure 8: Generated image samples from our ED2RM model and the DDPM baseline on the MNIST dataset (Sample 50).

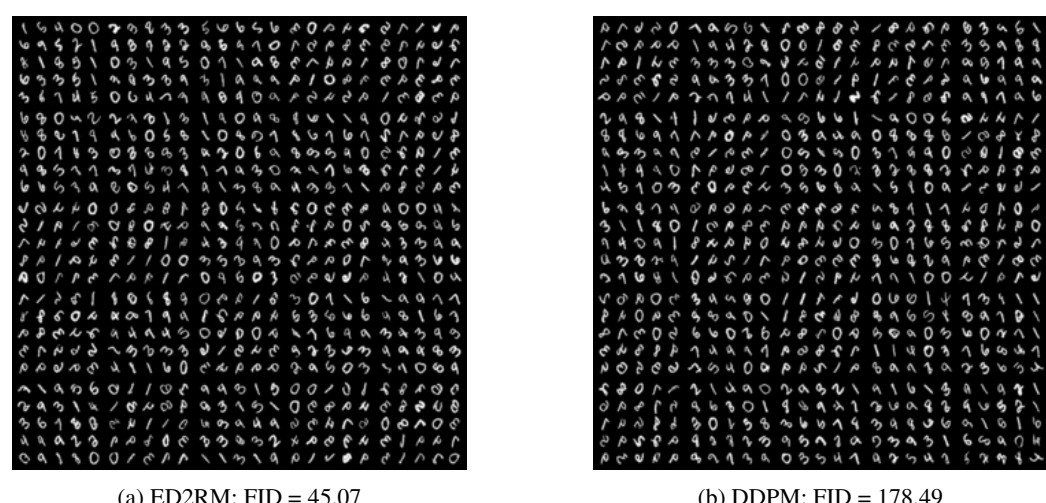

(a) ED2RM: FID = 45.07            (b) DDPM: FID = 178.49

Figure 9: Generated image samples from ED2RM and the standard DDPM on the RotoMNIST dataset(Sample 30).

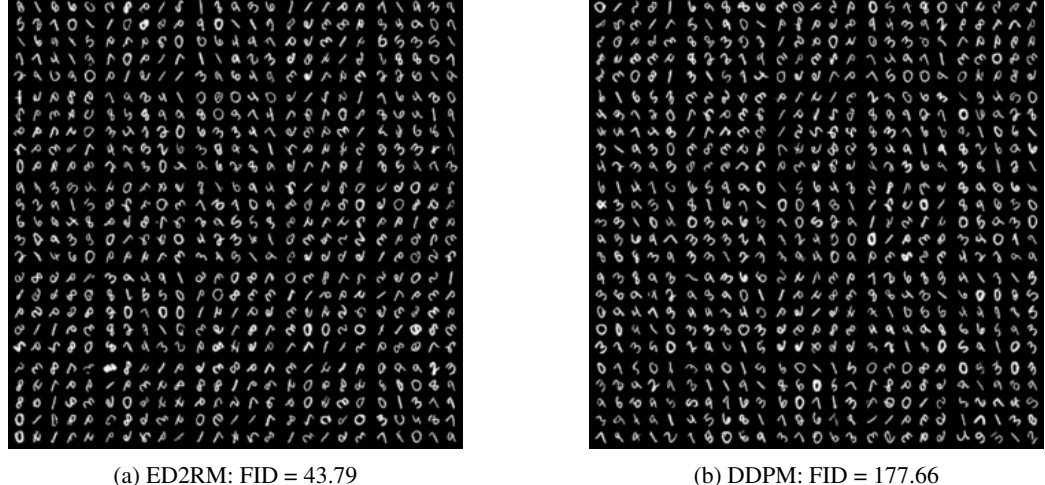

(a) ED2RM: FID = 43.79            (b) DDPM: FID = 177.66

Figure 10: Generated image samples from ED2RM and the standard DDPM on the RotoMNIST dataset (sample 50).

