# OpenReview forum: "ED2RM: Equivariant Denoising Diffusion Models based on Riemannian Morphological PDEs"
_ICLR.cc/2026/Conference — ICLR 2026 Conference Withdrawn Submission_

### Official Review · Reviewer_dYug · 2025-10-30

**Soundness:** 2
**Presentation:** 2
**Contribution:** 2
**Rating:** 4
**Confidence:** 4

**Summary:**

This paper introduces ED2RM, a framework that extends denoising diffusion probabilistic models (DDPMs) onto Riemannian manifolds through the integration of equivariant morphological PDE-G-CNN layers. The central idea is to impose E(n)-equivariance within the denoising network, by reformulating the learning dynamics in terms of geometrically interpretable partial differential equations (PDEs). Specifically, the authors replace the middle ResNet blocks of the standard diffusion U-Net with ResnetCDEBlocks, a module constructed from convection, dilation, and erosion PDEs operating on a Riemannian manifold. Empirical results on MNIST and RotoMNIST demonstrate that ED2RM achieves faster convergence and better FID scores than the vanilla DDPM.

**Strengths:**

1.	 The mathematical treatment is rigorous. The authors carefully derive the equivariant PDE operators (convection, dilation, and erosion), discuss their viscosity solutions, and show how they maintain equivariance under E(n) group actions on manifolds.
2.	The approach introduces a versatile way to inject symmetry priors and geometric structure into diffusion-based generative models. This could be influential in domains such as 3D shape synthesis, molecular graph generation, or medical image reconstruction.

**Weaknesses:**

1.	 Although the paper frequently emphasizes the importance of E(n)-equivariance, the Introduction lacks a concise conceptual explanation of what E(n) represents or why such symmetry should matter for diffusion processes. Without an introductory definition or example, readers unfamiliar with geometric deep learning may find it difficult to grasp the motivation behind introducing E(n)-equivariant structures so early in the paper
2.	 The Introduction lacks a clear rationale for why integrating equivariant PDE-G-CNNs into diffusion models is necessary or advantageous. Although previous studies have explored both equivariant diffusion models and PDE-G-CNNs, the paper does not specify what limitations of DDPMs this approach addresses or what unique benefits it offers. Consequently, the contribution appears as an intuitive architectural extension rather than a response to a defined gap.
3.	The paper only compares ED2RM with the standard DDPM, omitting stronger or more recent diffusion variants. A broader set of baselines would help clarify the relative merits of the approach.
4.	The data scale and diversity are limited. The paper conducts only preliminary experiments on MNIST and RotoMNIST, without evaluation on natural image datasets, which makes it difficult to comprehensively validate the effectiveness of the proposed method. Moreover, on the MNIST dataset, the performance of ED2RM is comparable to that of the baseline DDPM, indicating that the improvement is relatively modest in simpler scenarios.
5.	The paper does not include ablation studies to disentangle whether improvements stem from the equivariant design, the PDE-based layers, or the Riemannian manifold formulation. Without such analyses, it remains unclear which component drives the reported gains.

**Questions:**

1.	Could you elaborate on why DDPMs in particular benefit from PDE-G-CNN integration?
2.	How sensitive is the method to the choice of manifold (e.g., hyperbolic vs. spherical geometry)? Could the same principle generalize to other Riemannian structures?
3.	Have you evaluated or estimated the computational cost introduced by the PDE layers compared to regular convolutional ones?

---

### Official Review · Reviewer_1hxg · 2025-10-31

**Soundness:** 2
**Presentation:** 1
**Contribution:** 2
**Rating:** 2
**Confidence:** 2

**Summary:**

This work enhances denoising diffusion probabilistic models (DDPMs) by introducing PDE-G-CNNs, a geometric framework that integrates equivariant morphological partial differential equations into the prediction network. Formulated on Riemannian manifolds, the authors claim that the proposed approach improves geometric feature extraction and captures fine structural details.

**Strengths:**

Constructing diffusion models that incorporate additional geometric structures is an intriguing and valuable research direction worthy of further exploration.

**Weaknesses:**

As I am not quite familiar with the mathematical framework employed in this paper, I do not feel confident in reliably assessing the significance of its theoretical results. Therefore, my comments are primarily offered from a general machine learning perspective.

1. The paper is not well written, as it lacks clear explanations and intuitive motivations when introducing the theoretical components. This makes it difficult for researchers outside the specific area to grasp the main message of the work.
2. It is not clear to me what the benefit of this new diffusion model architecture is. What new features can this architecture achieve?
3. The empirical study is not convincing. In particular, the empirical study is performed on MNIST and ROTOMNIST. For these datasets, using FID is not reasonable, as it is not designed to handle non-natural images. In addition, authors should at least show some results on CIFAR-10 and/or CelebA to corroborate their claims.
4. I believe that improving sample quality should not be the primary motivation for incorporating additional geometric structures into the network. Such modifications are often computationally expensive and may negatively impact the model’s performance in other aspects. Instead, the authors could focus on specialized tasks where sampling inherently requires certain geometric features. Framing the work in this context would provide stronger motivation and make its significance easier to justify.

**Questions:**

Please refer to the weaknesses.

---

### Official Review · Reviewer_w7qz · 2025-11-05

**Soundness:** 3
**Presentation:** 2
**Contribution:** 2
**Rating:** 4
**Confidence:** 2

**Summary:**

This paper introduces ED2RM, an equivariant denoising diffusion model that tackles the poor handling of geometry and symmetries in standard generative models. Its key innovation is integrating Partial Differential Equations (PDEs) defined on a Riemannian manifold, which correspond to morphological operations like erosion and dilation. This principled design inherently guarantees equivariance to geometric transformations such as rotations and translations. Consequently, ED2RM demonstrates vastly superior robustness and generation quality compared to baseline DDPMs, particularly on datasets with significant geometric variations like RotoMNIST.

**Strengths:**

- The paper presents a interesting approach by deeply integrating principles from Riemannian geometry and PDEs into diffusion models. It guarantees geometric equivariance by design, rather than learning it from augmented data.
- The model's superiority is convincingly demonstrated through experimental results on rotation-heavy datasets like RotoMNIST.
- The framework holds significant potential for scientific applications requiring precise geometric modeling, such as in molecular design or 3D shape generation.

**Weaknesses:**

- The method's effectiveness has only been validated on simple datasets, leaving its scalability to complex, high-resolution images unproven.
- The model's reliance on Riemannian geometry and PDE solvers may entail significant computational overhead, especially when scaling to higher dimensions.
- The model's power stems from its equivariance to a specific, hand-designed Lie group. While highly effective for structured data, this imposes a strong inductive bias. Real-world processes involve complex, non-rigid and dynamic transformations (e.g., a dancer dancing, a butterfly flying) that cannot be neatly captured by such groups. This strong bias could paradoxically become a limitation, restricting the model's ability to learn the more plastic, deformable, and nuanced nature of real-world objects in motion.

---

If the authors can address my concerns, especially on how it would work in real-world scenarios, whether logically or experimentally, I would consider giving a higher score.

**Questions:**

- On Scalability and Computational Cost: How do you foresee the computational demands of the PDE solvers on the hyperbolic manifold scaling to high-dimensional data, such as megapixel color images? What are the primary bottlenecks, and are there potential approximations to make it feasible for larger-scale applications?
- On the Generality of Symmetries: The framework was designed to be equivariant to the Euclidean group E(n). How adaptable is this PDE-based approach to other, non-isomorphic symmetry groups, such as affine transformations or projective groups, which are relevant in other computer vision tasks?

**Details Of Ethics Concerns:**

None.

---

### Official Review · Reviewer_eUcq · 2025-11-05

**Soundness:** 2
**Presentation:** 1
**Contribution:** 2
**Rating:** 2
**Confidence:** 3

**Summary:**

This paper proposes a geometric approach to diffusion models by designing equivariant morphological partial differential equations (PDEs) for group convolutional neural networks (PDE-G-CNNs). Aiming to improve geometric feature extraction and equivariance in existing diffusion models, the method introduces a system of two PDEs: a convection term and a first-order Hamilton–Jacobi-type PDE that performs morphological multiscale dilations and erosions. Empirical validation on MNIST and RotoMNIST is provided to support the effectiveness of the proposed framework.

**Strengths:**

* The integration of morphological PDEs within a diffusion framework to preserve data symmetries throughout learning is novel and conceptually interesting.

**Weaknesses:**

* Methodological Motivation and Clarity:
  * A detailed comparison between ED2RM and prior Riemannian diffusion models is missing, making it difficult to discern the specific advancements contributed by this work.
  * The motivation for incorporating Riemannian manifolds and a PDE-based framework is not sufficiently justified, either through theoretical analysis or empirical ablation.
  * The logical connection between the definitions and propositions in Sections 3.1–3.2 and the actual model implementation in Section 3.3 is unclear, hindering reproducibility and understanding.
* Insufficient Experimental Validation:
  * Although the model is claimed to be equivariant to translations, rotations, reflections, and permutations, experimental evaluation is limited to simple image datasets with explicit rotation transformations (e.g., RotoMNIST). Broader equivariance claims lack empirical support.
  * There is no comparison to recent diffusion models that use equivariant score-based architectures, making it difficult to assess the relative benefit of the proposed PDE-based approach.

**Questions:**

* Beyond the group transformations mentioned in lines 84–85, can ED2RM be extended to other Lie groups? What are the limitations or requirements for such extensions?
* Could the authors provide further intuition behind the incorporation of Riemannian manifolds and PDE frameworks? What specific modeling limitations do these address compared to standard diffusion backbones?
* Could the authors provide more implementation details regarding the FID calculation on RotoMNIST?

---

### Official Review · Reviewer_DJ3a · 2025-11-05

**Soundness:** 3
**Presentation:** 3
**Contribution:** 2
**Rating:** 4
**Confidence:** 3

**Summary:**

This paper introduces an Equivariant Denoising Diffusion Riemannian Model (ED2RM) based on Riemannian morphological partial differential equations (PDEs). By incorporating convection terms and Hamilton-Jacobi equations into group convolutional neural networks (G-CNNs), the proposed framework addresses the limitations of conventional denoising diffusion probabilistic models in geometric feature extraction and equivariance preservation. Experimental results on MNIST and RotoMNIST generation tasks demonstrate that ED2RM achieves superior performance while maintaining enhanced equivariance properties.

**Strengths:**

Originality:
- This work presents a novel approach by investigating diffusion models within the framework of Riemannian manifolds and introducing convection, dilation, and erosion terms to maintain equivariance. The methodology developed for the diffusion model represents an original contribution to the field.

Quality:
- The introduction and literature review are thorough and well-structured.
- Numerical experiments partially validate both the effectiveness of the proposed method and the underlying theoretical foundations.

Clarity:
- The design of the new architectural components is clearly motivated and supported by theoretical reasoning.

Significance:
- The development of equivariance-preserving diffusion models is of considerable importance for specific generation tasks, such as molecular design, rendering the problem addressed in this study highly significant.

**Weaknesses:**

- Experiments:
  - The numerical experiments are currently limited to the MNIST dataset. It would be beneficial to evaluate the proposed method on additional benchmark datasets such as CIFAR-10, CelebA, or FFHQ to further demonstrate its generalizability.
  - An ablation study is lacking, making it difficult to assess the individual contributions of the convection, dilation, and erosion terms to the overall model performance.
  - The computational cost of the proposed method should be analyzed. Given that the introduced CDEBlocks involve solving convection and Hamilton–Jacobi equations, it is necessary to quantify any additional computational overhead incurred.
- Representation:
  - Terminology should be standardized throughout the manuscript. For instance, the rotated MNIST dataset is referred to inconsistently as both “ROTOMNIST” and “RotoMNIST.”
  - Inconsistencies appear in the reported results. For example, the values presented in Figure 2 and Table 1 do not align and should be cross-verified.
- Typos:
  - In Table 1, the FID score for ED2RM on RotoMNIST is listed as “49,30”; this should be corrected to “49.30” in accordance with standard numerical notation.

**Questions:**

- Could the authors please provide a reference or detailed description of the RotoMNIST dataset's construction methodology? Clarifying its provenance and generation process would enhance reproducibility.
- In Figure 2, why do several generated MNIST samples appear to exhibit mirroring artifacts? There are absolutely no samples like these in the original MNIST dataset.
- Definition 2.2 requires additional mathematical clarification. Could the authors please explicitly define the composition operator $(\phi \circ \mathcal{L}_{h}) f$, particularly regarding the functional relationships and domain mappings?

---

### Note · Authors · 2025-11-20

I have read and agree with the venue's withdrawal policy on behalf of myself and my co-authors.